# Exome sequencing in 16 patients with pituitary stalk interruption syndrome: A monocentric study

**Raja Brauner** [1]*, **Joelle Bignon-Topalovic**[2], **Anu Bashamboo**[2], **Ken McElreavey**[2]

**1** Pediatric Endocrinology Unit, Hôpital Fondation Adolphe de Rothschild and Université Paris Cité, Paris, France, **2** Human Developmental Genetic Unit, Institut Pasteur, Paris, France

* raja.brauner@wanadoo.fr

**Data Availability Statement:** All relevant data are within the manuscript and all novel variants are deposited with ClinVar (https://www.ncbi.nlm.nih.

## Abstract

Pituitary stalk interruption syndrome (PSIS) is a rare disorder characterized by an absent or ectopic posterior pituitary, absent or interrupted pituitary stalk and anterior pituitary hypoplasia on magnetic resonance imaging (MRI), as well in some cases a range of heterogeneous somatic anomalies. The triad can be incomplete. Here, we performed exome sequencing on 16 sporadic patients, aged 0.4 to 13.7 years diagnosed with isolated or complex PSIS. Growth hormone deficiency was isolated in 10 cases, or associated with thyrotropin deficiency in 6 others (isolated (2 cases), associated with adrenocorticotropin deficiency (1 case), gonadotropins deficiency (1 case), or multiple deficiencies (2 cases)). Additional phenotypic anomalies were present in six cases (37.5%) including four with ophthalmic disorders. In 13 patients variants were identified that may contribute to the phenotype. However, only a single individual carried a variant classified as pathogenic. This child presented with the typical clinical presentation of Okur-Chung neurodevelopmental syndrome due to a *CSNK2A1* missense variant. We also identified variants in the holoprosencephaly associated genes *GLI2* and *PTCH1*. A likely pathogenic novel splice site variant in the *GLI2* gene was observed in a child with PSIS and megacisterna magna. In the remaining 11 cases 26 variants in genes associated with pituitary development or function were identified and were classified of unknown significance. Compared with syndromic forms the diagnostic yield in the isolated forms of PSIS is low. Although we identified rare or novel missense variants in several hypogonadotropic hypogonadism genes (e.g. *FGF17*, *HS6ST1*, *KISS1R*, *CHD7*, *IL17RD*) definitively linking them to the PSIS phenotype is premature. A major challenge remains to identify pathogenic variants in cases with isolated PSIS.

## Introduction

Pituitary stalk interruption syndrome (PSIS) is a rare disorder characterized by the combination of specific findings in magnetic resonance imaging (MRI) including an absent or ectopic posterior pituitary, absent or interrupted pituitary stalk and anterior pituitary hypoplasia [1]. The triad can be incomplete. The consequences of PSIS are a series of anterior pituitary

gov/clinvar/) using accession codes VCV003852768, SCV000978577, SCV000978576, VCV000978570 or at the LOVD system at Global Variome Ltd (https://www.lovd.nl) with variant ID 0000933708.

**Funding:** The authors received no specific funding for this work.

**Competing interests:** The authors have declared that no competing interests exist.

deficiencies including growth hormone (GH) deficiency that may be isolated or associated with other hormonal deficiencies including thyroid-stimulating hormone (TSH), adrenocorticotropin (ACTH) and/or luteinising hormone/follicle stimulating hormone (hypogonadotropic hypogonadism, HH). Posterior pituitary deficiency leading to central diabetes insipidus is very rare in PSIS.

Pathogenic or potentially pathogenic variants have been reported in PSIS associated with a wide range of genes involved in pituitary development or function [2–8] as well as genes associated with holoprosencephaly [9–11]. We previously analysed by exome sequencing 52 patients monitored for PSIS by the same pediatric endocrinologist and identified pathogenic and potentially pathogenic mutations in 39 patients [12]. Many of these cases where pathogenic variants were identified presented with complex phenotypes in addition to PSIS including seizures, neurodevelopmental disability, micropenis or cryptorchidism, which are usually considered as secondary to the pituitary deficiencies. However, we demonstrated they are likely due to pathogenic variants responsible for epilepsy, cerebral or cerebella development, thyroid development or HH. This suggests that pathogenic variants are more likely to be identified in PSIS associated with other known genetic syndromes rather than as specific clinical entity.

Here, we analysed by exome sequencing 16 new patients diagnosed with PSIS. We identified pathogenic or likely pathogenic variants in only two patients, one presented with classical Okur-Chung neurodevelopmental syndrome and the other with isolated PSIS. Most of the patients in this series did not present with anomalies other than PSIS. Together with published data this suggests that obtaining a genetic diagnosis in cases of isolated PSIS will remain challenging. This is due to both the rarity of the phenotype and lack of familial cases which hinders formal genetic analysis as well as the lack of knowledge on the genetic pathways involved in pituitary formation and development.

## Materials and methods

### Patients

The diagnosis of PSIS was made on MRI if the patient present 1) an ectopic or absent posterior pituitary gland and/or 2) an interrupted, not seen, thin or malformed pituitary stalk, with or without hypoplasia or aplasia of the anterior pituitary gland. All patients had an ectopic posterior pituitary gland on MRI, except two patients where it was not seen and one with posterior pituitary gland in normal position but interrupted pituitary stalk (case 7). Thus, the pituitary stalk was defined as interrupted (n = 5), not seen (n = 5), thin (n = 3), malformed (n = 1) and normal (n = 2). The anterior pituitary gland height was measured on the midline T1 weighted scan [1]. It was low (n = 9), normal at 4 mm (n = 5) and not measured (n = 2).

This retrospective single-center study was performed in 16 individuals (13 boys and 3 girls). Thus, among the 98 PSIS patients monitored for hypothalamic-pituitary deficiency by a senior pediatric endocrinologist (R. Brauner) in a university hospital between 1978 and 2021 (43 years), 68 have a DNA sample available for ES. The results of ES of 52 of them were previously reported [12]. The data were accessed for the present work between January 2020 and December 2022.

### Methods

The pituitary evaluation and follow-up were conducted as previously described [13]. Hypoglycemia was defined as a blood glucose concentration below 3 mmol/L after 2 days of age. Decreased growth rate was defined as a height velocity during the previous year of more than one standard deviation score below the mean for chronological age or decrease in height

standard deviation of more than 0.5 over 1 year in children older than 2 years. Micropenis was defined as a penis length of less than 30 mm.

The criterion for diagnosing GH deficiency was a GH peak response of less than 20 mU/L or 6.7 ng/mL after two pharmacological stimulation tests or during spontaneous hypoglycemia, excluding the response to GH-releasing hormone, with low insulin-like growth factor 1 concentration. As we used various GH assays over the study period, we expressed the GH peak concentration in mU/L using conversion factors (ng/mL to mU/L) that were specific of the international standard used to calibrate the GH assay. TSH deficiency was diagnosed by plasma free thyroxin below 12 pmol/L. ACTH deficiency was diagnosed by basal plasma cortisol concentrations at 8 a.m. below 40 ng/mL (110 nmol/L) in neonates and below 80 ng/mL (220 nmol/L) in older children, with no increase during hypoglycemia and low/normal ACTH concentration. HH was diagnosed by the absence of pubertal development at 13 years in girls and 14 years in boys and no or partial gonadotropins response to a gonadotropin-releasing hormone stimulation test [14]. The follow-up for each patient included measurements of plasma free thyroxin and cortisol concentrations at 8 a.m. every one or two years, if their concentrations had previously been normal to diagnose delayed deficiency.

### Exome sequencing and array-CGH analysis

Exon enrichment was performed as described elsewhere using Agilent SureSelect Human All Exon V4 [15]. Paired-end sequencing was performed on the Illumina HiSeq2000 platform with an average sequencing coverage of x50. Read files were generated from the sequencing platform via the manufacturer's proprietary software. Reads were mapped using the Burrows–Wheeler Aligner and local realignment of the mapped reads around potential insertion/deletion (indel) sites was carried out with the GATK version 1.6. SNP and indel variants were called using the GATK Unified Genotyper for each sample. SNP novelty was determined against dbSNP138. Datasets were filtered for novel or rare (MAF<0.01) variants. Novel and rare variants were analyzed by a range of web-based bioinformatics tools using the EnsEMBL SNP Effect Predictor (http://www.ensembl.org/homosapiens/userdata/uploadvariations). All variants were screened manually against the Human Gene Mutation Database Professional (Biobase) (http://www.biobase-international.com/product/hgmd). *In silico* analysis was performed to determine the potential pathogenicity of the variants using Polyphen (http://genetics.bwh.harvard.edu/pph), and SIFT (http://sift.jcvi.org/www/SIFT_chr_coords_submit.html) online tools that predict the effect of human mutations on protein function. We focused our analyses on non-synonymous coding, nonsense, and splice site variants, filtering out all known common variations contained in dbSNP (build 138) (www.ncbi.nlm.nih.gov/projects/SNP/), and in the gnomAD database (http://gnomad.broadinstitute.org/). An in-house database of 1100 exomes from control individuals or individuals with unrelated pathologies were also screened for the potential pathogenic variants identified in the PSIS cohort. Variants were confirmed by visual examination using the IGV browser or by Sanger sequencing. Variants were classified according to ACMG guidelines [16]. The parents' DNA was unavailable for study, therefore trio analysis was not possible. Exome datasets were also compared to an in-house control dataset of >700 exomes. Karyotyping was performed using standard methods and chromosomes were observed after G and R banding.

### Results and discussion

The age at diagnosis ranged from 0.4 to 13.7 years (Table 1). There was no consanguinity nor familial forms of PSIS. The initial symptom leading to the diagnosis of PSIS was hypoglycemia in the youngest patient (case 1), jaundice and ophthalmic symptom in two (cases 2 and 10),

**Table 1. Phenotypes of 16 patients with PSIS analysed by exome sequencing.**

| Case | Ethnic Origin | Sex | Age at diagnosis (y) | Initial symptom | Pituitary deficits associated with GH deficiency | Associated phenotypes | MRI Peculiar features | PP | Stalk | Pituitary height, mm |
|---|---|---|---|---|---|---|---|---|---|---|
| 1 | France/ Italy | M | 0.4* | Hypoglycemia | T, HH | Fanconi, microphtalmia, moderate hepatic fibrosis, Factor V Leiden mutation | | E | not seen | 1 |
| 2 | Asia/ France | M | 0.5 | Neonatal jaundice, ophtalmic | None, prepubertal | L optic atrophia, coloboma | 2 posterior ectopic pituitaries | E | N | 2 |
| 3 | France | M | 1.6 | Decreased GR | None, N puberty | | 2nd anterior pituitary replacing the pituitary stalk | not seen | malformed | 2.7 |
| 4 | France | M | 2.6 | Decreased GR | None, prepubertal | Neurodevelopmental syndrome | | not seen | N | 4 |
| 5 | France | M | 3.6 | Decreased GR | None, N puberty | | | E | I | 2 |
| 6 | Portugal | M | 4.3 | Decreased GR | None, N puberty | | | E | thin | NA |
| 7 | France | F | 4.5 | Decreased GR | None, early and rapid puberty | | | N | I | 4 |
| 8 | France | M | 4.7 | Decreased GR | None, prepubertal | | | E | not seen | 4 |
| 9 | France | F | 5.7 | Decreased GR | T, C**, HH | L strabismus convergent, unilateral amblyopia, no anosmia | | E | not seen | 4 |
| 10 | Marocco | F | 6.2 | Neonatal jaundice, strabismus | T, C**, N puberty | Septo optic dysplasia | Septum pellucidum, L hypoplasia optic chiasma | E | thin | small |
| 11 | Portugal | M | 6.8 | Decreased GR | None, prepubertal | | | E | I | 3 |
| 12 | France | M | 8.9 | Decreased GR | None, N puberty | | | E | thin | 4 |
| 13 | Tunisia | M | 9.2 | Decreased GR | T, C**, HH | | | E | I | NA |
| 14 | France | M | 10.4 | Decreased GR | T, N puberty | | | E | not seen | 2.2 |
| 15 | France | M | 10.6* | Decreased GR | T, prepubertal | Fanconi, nephroblastoma, leukemia and cancer mother | Brainstem hypoplasia | E | not seen | 1 |
| 16 | France | M | 13.7 | Decreased GR | None, N puberty | | Megacisterna magna | E | I | 2 |

Abbreviations: Deficits in GH growth hormone, T thyrotropin, C adrenocorticotropic; HH hypogonadotropic hypogonadism;

GR: growth rate; L: left; N: normal; PP posterior pituitary; E ectopic; I interrupted

*Micropenis, cryptorchidism

**Cortisol at 8 a.m. at 64, 10.7 and 15 ng/mL in cases 9, 10 and 13 respectively

and decreased growth rate in the 13 others. The GH deficiency was isolated in 10 cases (62.5%), or associated with TSH deficiency isolated in two cases (12.5%) or associated with ACTH deficiency in one case (6.25%) or with HH in one case (6.25%), or multiple deficiencies in two (12.5%). Thus eight patients had normal spontaneous puberty, including a girl who had early and rapid puberty (case 7), 5 cases were in prepubertal age and three cases had HH (18.75% or 27.27% after excluding 5 in prepubertal age). No patient had diabetes insipidus. The associated phenotypes (6 cases, 37.5%) are detailed in the Table 1. Ophthalmic anomalies are present in four patients and Fanconi anemia in two [17].

The genetic cause of most individuals presenting with PSIS with or without associated pituitary or other anomalies is unknown [12, 18]. Both single gene anomalies as well as more complex multigenic inheritance have been proposed to explain the phenotype [19, 20]. The genomic study of this cohort reflects this with pathogenic or likely pathogenic variants identified in only two of the 16 cases suggesting that either multigenic inheritance is involved or that variants in non-coding sequences play an important role (Table 2). The chromosome

**Table 2. Gene variants identified in 13 individuals with PSIS that may be associated with the phenotype.**

| Case | Gene | Variant/dbSNP /Zygosity/SIFT/Polyphen2 | MAF and population | ClinVar No /Interpretation | MI | Associated phenotypes (MIM) | ACMG classification |
|------|------|------|------|------|------|------|------|
| 1 | FANCD2 | NM_033084:c.A782T:p.K261M/ rs778289599/ heterozygous/SIFT-D/Polyphen2-P | 0.00002891 Latino/ Admixed American | 929646 Pathogenic | AR | Fanconi anemia, complementation group D2 (227646) | VUS |
| | DNMT1 | NM_001379:c.C406T:p.R136C/ rs138841970/ heterozygous/SIFT-T/Polyphen2-B | 0.0004258 Europen non-finnish | 234461 Conflicting | AD | Cerebellar ataxia, deafness, and narcolepsy, autosomal dominant (604121); Neuropathy, hereditary sensory, type IE (614116) | VUS |
| | RFWD3 | NM_018124: c.A1364G: p.Y455C/rs200354694/ heterozygous/SIFT-D/Polyphen2-D | 0.0002769 Other | - | AR | Fanconi anemia, complementation group W (617784) | VUS |
| 2 | DMXL2 | NM_015263: c.4485_4486insTATTTCTATTAGGTACAACTTT: p.I1496_N1497delinsYFYX/-/heterozygous | Novel | - | AR | Developmental and epileptic encephalopathy 81 (618663) Deafness autosomal dominnt 71 (617605) Polyendocrine-polyneuropathy syndrome (616113) | VUS |
| | TMEM67 | NM_001142301:c.C1843A:p.L615I/-/heterozygous/ SIFT-T/Polyphen2-B | Novel | - | AR | COACH syndrome (216360); Joubert syndrome 6 (610688); Meckel syndrome 3 (607361); Nephronophtisis 11 (613550) | VUS |
| | TMEM67 | NM_001142301: /-/heterozygous | Novel | - | AR | COACH syndrome (216360); Joubert syndrome 6 (610688); Meckel syndrome 3 (607361); Nephronophtisis 11 (613550) | VUS |
| 3 | GLI2 | NM_005270:c.1368+1G>A/-/heterozygous | Novel | - | AD | Culler-Jones syndrome (615849); Holoprosencephaly 9 (610829) | Likely pathogenic |
| 4 | LHX9 | NM_020204:c.C1063A:p.P355T/-/heterozygous/ SIFT-D/Polyphen2-PrD | Novel | - | - | - | VUS |
| | CSNK2A1 | NM_177559:c.A593G:p.K198R/ rs869312840/ heterozygous/SIFT-T/Polyphen2-PrD | 0.000008798 European Non-Finnish | 224790 Conflicting interpretations of pathogenicity Pathogenic; Likely pathogenic; Uncertain significance | AD | Okur-Chung neurodevelopmental syndrome (617062) | Pathogenic |
| 5 | GLI3 | NM_000168:c.A2424G:p.I808M/ rs62622373/ heterozygous/SIFT-T/Polyphen2-D | 0.009359 Ashkenazi Jewish | 235210 LB | AD | Greig cephalopolysyndactyly syndrome (175700); Pallister-Hall syndrome (146510); Polydactyly, postaxial, types A1 and B (174200); Polydactyly, preaxial, type IV (174700) | VUS |
| 7 | PITX1 | NM_002653:c.403-1->CCGAGCCGCGCGTGCG/-/ heterozygous | Novel | - | AD | Clubfoot, congenital, with or without deficiency of long bones and/or mirror-image polydactyly (119800); Liebenberg syndrome (186550) | VUS |
| | PITX2 | NM_001204397:c.C501A:p.Y167X/-/heterozygous | Novel | - | AD | Anterior segment dysgenesis 4 (137600); Axenfeld-Rieger syndrome, type 1 (180500); Ring dermoid of cornea (180550) | VUS |
| 9 | CC2D2A | NM_001080522:c.C3055T:p.R1019X/ rs370880399/ heterozygous/ | 0.001386 Ashkenazi Jewish | 217602: Pathogenic/likely pathogenic | AR | COACH syndrome (216360); Joubert syndrome 9 (612285) Meckel syndrome (612284) | VUS |
| | HESX1 | NM_003865:c.G200C:p.S67T/ rs141863326/ heterozygous/SIFT-T/Polyphen2-B | 0.0006206 South Asian | 289279 Uncertain | AD/ AR | Pituitary hormone deficiency, combined, 5 (182230) | VUS |

(*Continued*)

**Table 2.** (Continued)

| Case | Gene | Variant/dbSNP /Zygosity/SIFT/Polyphen2 | MAF and population | ClinVar No /Interpretation | MI | Associated phenotypes (MIM) | ACMG classification |
|------|------|----------------------------------------|--------------------|---------------------------|-----|----------------------------|---------------------|
|  | *PTCH1* | NM_001083607:c.G2032A:p.V678M/ rs201125580/ heterozygous/SIFT-D/Polyphen2-PrD | 0.004051 Ashkenazi Jewish | 41655 Conflicting | AD | Holoprosencephaly 7 (610828) | VUS |
|  | *HS6ST1* | NM_004807:c.C652T:p.P218S/ rs200268730/ heterozygous/SIFT-T/Polyphen2-PrD | 0.003100 European Non-Finnish | 1297575 Conflicting | AD | Hypogonadotropic hypogonadism 15 with or without anosmia (614880) | VUS |
| 10 | *CHD7* | NM_017780:c.C8416G:p.L2806V/ Rs45521933/ heterozygous/SIFT-T/Polyphen2-P | 0.004017 Latino/ Admixed American | 95814 LB/B | AD | CHARGE syndrome (214800); Hypogonadotropic hypogonadism 5 with or without anosmia (612370) | VUS |
|  | *IL17RD* | NM_017563:c.C470T:p.T157M/ rs200321574/ heterozygous/SIFT-D/Polyphen2-D | 0.0001387 Other | - | AR/ AD | Hypogonadotropic hypogonadism 18 with or without anosmia (606807) | VUS |
|  | ARID1A | NM_139135:exon1:c.G376A:p.G126S/ rs1400214289/heterozygous/SIFT-D/Polyphen2-PD | 0.0001971 Latino/ Admixed American | 1197625 LB | AD | Coffin-Siris syndrome 2 (614607) | VUS |
| 11 | *TBCE* | NM_003193:c.C89T:p.P30L/-/heterozygous/ SIFT-D/Polyphen2-D | Novel | - | AR | Encephalopathy progressive, with amyotrophy and optic atrophy (617207); Hypoparathyroidism-retardation-dysmorphism syndrome (241410); Kenny-Caffey syndrome type 1 (244460) | VUS |
| 12 | *ARNT2* | NM_014862:c.G1707T:p.Q569H/ rs145379118/ heterozygous/SIFT-D/Polyphen2-B | 0.005086 European Non-Finnish | 791642 LB | AR | Webb-Dattani syndrome (615926) | VUS |
| 13 | *PROP1* | NM_006261:c.C425T:p.A142V/ rs143790367/ heterozygous/SIFT-T/Polyphen2-B | 0.006080 Ashkenazi Jewish | 196432 Conflicting | AR | Pituitary hormone deficiency, combined, 2 (262600) | VUS |
|  | *KISS1R* | NM_032551:c.G565A:p.A189T/ rs73507527/ heterozygous/ SIFT-T/Polyphen2-B | 0.06569 African/ African Americans | 235657 B/LB | AR | Hypogonadotropic hypogonadism 8 with or without anosmia (614837) | VUS |
|  | *GLI2* | NM_005270:c.G2159A:p.R720H/ rs149091975/ heterozygous | 0.001366 African/ African American | 259723 Uncertain significance | AD | Culler-Jones syndrome (615849); Holoprosencephaly 9 (610829) | VUS |
| 14 | *DMXL2* | NM_015263:c.88-2->TTTTTTTTTTTT/-/ homozygous | Novel | - | AR | Developmental and epileptic encephalopathy 81 (618663) Deafness autosomal dominnt 71 (617605) Polyendocrine-polyneuropathy sd (616113) | VUS |
| 16 | *GLI2* | GLI2:NM_005270:exon12:c.G2041A:p.V681M/ rs551617843/SIFT-D/Polyphen2-B | 0.0003673 Latino/ Admixed American | 1411228 uncertain | AD | Culler-Jones syndrome (615849); Holoprosencephaly 9 (610829) | VUS |
|  | PTCH1 | NM_001083607:c.A3494G:p.Y1165C/rs147067171/ heterozygous/SIFT-T/Polyphen2-D | 0.001075 Ashkenazi Jewish | 132723 Conflicting (B) | AD | Holoprosencephaly 7 (610828) | VUS |
|  | PTCH1 | NM_001083607:c.C3871T:p.R1291W/rs143464326/ heterozygous/SIFT-T/Polyphen2-B | 0.005945 South Asian | 41666 Conflicting | AD | Holoprosencephaly 7 (610828) | VUS |

(*Continued*)

**Table 2.** (Continued)

| Case | Gene | Variant/dbSNP /Zygosity/SIFT/Polyphen2 | MAF and population | ClinVar No /Interpretation | MI | Associated phenotypes (MIM) | ACMG classification |
|------|------|----------------------------------------|--------------------|----------------------------|----|-----------------------------|---------------------|
|      | FGF17 | NM_003867:c.C634T:p.P212S/rs778556940/ heterozygous/SIFT-T/Polyphen2-PoB | 0.000209 Latino/ Admixed American | - | AD | Hypogonadotropic hypogonadism 20 with or without anosmia (615270) | VUS |

MAF, Minor Allelic Frequency (the population showing the highest MAF is indicated); MI, known mode of inheritance of the phenotype; AR, autosomal recessive; AD, autosomal dominant; T, tolerated; B, benign; PoB probably benign; LB Likely benign; D, deleterious; PrD probably damaging; VUS, variant of unknown significance

complement of all individuals in the study was found to be normal following standard karyotype analysis.

In 13 patients variants were identified that may contribute to the phenotype. Genetic variants associated with the phenotype could not be identified in candidate genes in the cases 6,8 and 15. In only one of patient was a variant classified as pathogenic. This child (case 4) presented with the typical clinical presentation of Okur-Chung neurodevelopmental syndrome with delay in walk (2.4 years), hypotonia and peculiar form of autism. This is a very rare autosomal dominant developmental disorder characterized by neurological anomalies with variable dysmorphic features, hypotonia, skeletal anomalies and cardiac anomalies. The phenotype is highly variable including brain MRI while others do not exhibit any abnormality in the brain architecture [21]. Okur-Chung neurodevelopmental syndrome is caused by *de novo* variants in the *CSNK2A1* gene, which encodes the alpha subunit of protein kinase CK2, a highly conserved serine/threonine protein kinase that phosphorylates a large number of substrates containing acidic residues C-terminal to the phosphorylated serine or threonine [22]. The p.K198R variant is a recurrent mutation with at least eight other cases reported in the literature with this pathogenic variant [23–27]. The K198 residue is located within the kinase domain and specifically within the activation segment of CK2 [28]. Although p. K198R was originally considered to be a loss-of-function variant more recent studies have indicated that variant does not result in complete loss of kinase activity, but rather a shift in substrate specificity [28]. In this case presented here we do not have access to the parents' DNA to determine if the variant is *de novo* but based on previously published cases associated with this missense variant together with functional studies we classified this variant as pathogenic.

One of the main findings in recent years concerning the genetic factors involved in PSIS, and other pituitary anomalies, is the link to genes known to cause holoprosencephaly. This includes for example the genes *CDON*, *GLI2*, *SHH* and *TGIF1* [11, 12, 29]. In this cohort we identified several variants in the holoprosencephaly genes *PTCH1* and *GLI2* that may contribute to PSIS. However, in only one case (case 3) could the variant be classified as likely pathogenic. He has a peculiar aspect on MRI with second anterior pituitary like replacing the pituitary stalk. This child carried a novel and essential splice site variant that is predicted to be a loss-of-function (LOF) variant in the *GLI2* gene. Pathogenic, heterozygous variants in *GLI2* are associated with either autosomal dominant Culler-Jones Syndrome or holoprosencephaly 9 with considerable phenotypic heterogeneity [30]. The most common features associated with *GLI2* LOF variants are polydactyly, hypopituitarism (isolated or combined) and abnormal pituitary imaging. These features are observed in approximately 50% of cases and the polydactyly is often the mildest phenotypic expression and often absent or not reported in parents carrying known pathogenic variants transmitted to their more severely affected child [30].

Indeed, *GLI2* variants have also been associated with PSIS [12, 29, 31] and hence we consider this variant to be likely pathogenic.

The role of other holoprosencephaly gene variants that we identified is less clear. Patient 16 carries two variants in the *PTCH1* gene. He had megacisterna magna on his MRI. Pathogenic heterozygous variants in *PTCH1*, a gene that encodes the receptor for the secreted hedgehog ligands are responsible for an autosomal dominant form of holoprosencephaly that includes panhypopituitarism [32]. Heterozygous variants have also been proposed to cause PSIS but currently the evidence for pathogenicity is limited [29].

For the other variants identified in this study the relationship with the PSIS phenotype is more difficult to determine. Patient 7 carries two heterozygous, predicted loss-of-function variants in the *PITX1* and *PITX2* genes (Table 2). Variants in both of these genes are associated with complex autosomal dominant congenital disorders [33]. *PITX2* plays a role during ontogeny of the pituitary gland and other anterior structures, including the eye and patients with heterozygous pathogenic *PITX2* variants present with eye, pituitary craniofacial, dental, cardiac, and umbilical anomalies [33]. However, patient 7, despite carrying a nonsense variant in *PITX2* only presented with a decreased growth rate, early puberty and PSIS. No eye nor other pituitary anomalies were observed. Since the patient carries a novel loss-of-function nonsense variant, it is possible that this case represents a very mild expression of the *PITX2* phenotype.

Patient 1 was diagnosed with PSIS associated with Fanconi anemia. The child carried a known heterozygous pathogenic variant in *FANCD2*, however Fanconi anemia due to *FANCD2* variants is autosomal recessive [34]. It is therefore possible that the affected child carried a second variant, perhaps non-coding that was not covered by the exome sequencing, or the phenotype may be due to the combination of *FANCD2* with variants in the *DNMT1* or *RFWD3* genes (Table 2). Biallelic variants in the latter were reported in a familial form of Fanconi anemia [35]. The clinical description of patient reported by Knies et al., is similar to patient 1. They both presented with Fanconi anemia, intrauterine growth retardation, left kidney dysplasia, left radius hypoplasia, absent thumbs, GH deficiency, ectopic posterior pituitary and with no intellectual disability.

Many of the remaining variants were classified as variant of uncertain significance (VUS) either because the allelic frequency of the variant in the general population was too high to cause of the phenotype (e.g. KISS1R p.A189T), and/or the variant was monoallelic, where the disease is associated with biallelic variants (e.g. CC2D2A, p.1019X) and/or the variant was not predicted to be pathogenic by predictive software (e.g. KISS1R p.A189T). This includes variants in the genes associated with HH. Although we identified rare or novel missense variants in several HH genes (e.g. *FGF17*, *HS6ST1*, *KISS1R*, *CHD7*, *IL17RD*) [36] definitively linking them to the PSIS phenotype is premature.

## Conclusions

We previously reported that in many individuals PSIS may be considered as part of the phenotypic spectrum of other known genetic syndromes rather than as a specific clinical entity. In these individuals the genetic cause can often be explained as the genetic syndrome is well defined (e.g. FANCA) [12]. The major challenge is to identify pathogenic variants in cases with isolated PSIS. As shown here and elsewhere [18, 37, 38], in such individuals the diagnostic yield is currently very low.

## Author Contributions

**Conceptualization:** Raja Brauner, Ken McElreavey.

**Data curation:** Joelle Bignon-Topalovic, Anu Bashamboo.

**Investigation:** Raja Brauner.

**Writing – original draft:** Raja Brauner, Ken McElreavey.

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
