## [Decision Letter · Decision Letter 0]

16 Aug 2023

PONE-D-23-22676Exome sequencing in 16 patients with pituitary stalk interruption syndrome: a monocentric studyPLOS ONE

Dear Dr. Brauner,

Thank you for submitting your manuscript to PLOS ONE. After careful consideration, we feel that it has merit but does not fully meet PLOS ONE’s publication criteria as it currently stands. Therefore, we invite you to submit a revised version of the manuscript that addresses the points raised during the review process.

We look forward to receiving your revised manuscript.

Kind regards,

Nejat Mahdieh

Academic Editor

PLOS ONE

Reviewers' comments:

Reviewer's Responses to Questions

**Comments to the Author**

1. Is the manuscript technically sound, and do the data support the conclusions?

Reviewer #1: Yes

Reviewer #2: Yes

2. Has the statistical analysis been performed appropriately and rigorously? 

Reviewer #1: N/A

Reviewer #2: Yes

3. Have the authors made all data underlying the findings in their manuscript fully available?

Reviewer #1: Yes

Reviewer #2: Yes

4. Is the manuscript presented in an intelligible fashion and written in standard English?

Reviewer #1: Yes

Reviewer #2: Yes

5. Review Comments to the Author

Reviewer #1: The authors conducted exome sequencing on 16 unpublished PSIS cases, most of which were idiopathic, out of the 68 cases they followed up at their facility. As a result, they found that many of the identified variants were VUS, highlighting the challenges of genetic diagnosis in idiopathic PSIS.

This study conducted exome sequencing on rare diseases, and given the rarity of the disease, the small sample size of 16 is considered inevitable. However, the conclusion that "genetic diagnosis of idiopathic PSIS is challenging" might be too weak. Considering that the low diagnostic yield in idiopathic PSIS has been previously reported [Jullien N et al, Clin Endocrinol 94(2):277–289, 2020] as the authors cited, it might be difficult to define the scientific significance of the study. Is it possible to draw a conclusion that more strongly emphasizes the scientific significance of the study?

Minor comments are noted below.

What were the criteria used to diagnose PSIS? From Table 1 alone, the reviewer did not understand how the diagnosis of PSIS was made. The description about abbreviations in the footnote of Table 1 seems to be truncated. Also, it is unclear whether a pituitary height of 4 is small or not.

Reviewer #2: Authors performed exome sequencing in 16 patients with isolated PSIS and described variants they found. It is an interesting report, but I have some comments.

On what basis do the authors define the ACTH deficiency? Serum cortisol varies so much in individuals that it would be difficult to define it without a functional test. Also, hypoglycemia does not necessarily occur in ACTH deficiency. Since ACTH deficiency may be over or underestimated, a limitation should be stated.

How to evaluate the function of the VUS found and how the pathogenic mutation found is involved in the structural abnormality of the stalk should be mentioned in the discussion.

Table 1: Abbreviation section was interrupted. What does “E” and “I” mean? What is the unit of “pituitary height”? When the stalk is normal, how did authors diagnose PSIS?

Table 2: Abbreviation section was missing.

What does “MAF and population” mean? I assume that authors noted “minor allele frequency” and its given population. So, the SNP was not detected in other populations?

For the case 1, FANCD2, information of MAF and population were inversed.

What does “MI” mean? If it is “AD”, does it mean the SNP can induce the phenotype as autosomal dominant? I am wondering how those SNPs were recognized as dominant, as most of SNPs were “VUS”. If these “MI” means the genetic form of “MIM”, it is difficult for readers to understand.

Line 142:” No candidate gene was identified in cases 6,8 and 15.” Does it mean that “no mutation/SNP of the candidate gene”?

6. PLOS authors have the option to publish the peer review history of their article (what does this mean?). If published, this will include your full peer review and any attached files.

Reviewer #1: No

Reviewer #2: No

---

## [Author Response · Author response to Decision Letter 0]

24 Aug 2023

To Reviewer 1

Dear Reviewer,

Thank you for reviewing our manuscript.

As you suggested, the conclusions that “genetic diagnosis of idiopathic PSIS is challenging” of the Abstract and of the Introduction have been completed.

The criteria used to diagnose PSIS are now at the onset of “Patients”. The anterior pituitary gland height, mainly 4 mm, has been indicated. We added to the introduction of the Abstract that “The triad of PSIS can be incomplete”.

The abbreviations of Table 1 have been completed.

To Reviewer 2

Dear Reviewer,

Thank you for reviewing our manuscript.

Concerning the definition of the ACTH deficiency, the values indicated in “Methods” have been completed by the individual values of the 3 patients with ACTH deficiency (below Table 1).

Table 1: Abbreviation section has been completed and the unit of pituitary height added.

As suggested by the other Reviewer, we completed the diagnosis criteria of the PSIS and detailed the pituitary height at the onset of “Patients”. We added to the introduction of the Abstract that “The triad of PSIS can be incomplete”.

Q: What does “MAF and population” mean? I assume that authors noted “minor allele frequency” and its given population. So, the SNP was not detected in other populations?

A: MAF is the minor allelic frequency and it cited population is the population with the highest known frequency of the variant in GnomAD. The variant may well be present in other populations but at lower allelic frequencies.

Q: For the case 1, FANCD2, information of MAF and population were inversed.

A: This has been corrected.

Q: What does “MI” mean? 

A: MI is the known mode of inheritance of the disorder associated with the gene. It does not reflect the mode of inheritance in the particular case. 

Q: Line 142:” No candidate gene was identified in cases 6,8 and 15.” Does it mean that “no mutation/SNP of the candidate gene”?

A: For clarity this sentence has been modified to “Genetic variants associated with the phenotype could not be identified in candidate genes in the cases 6,8 and 15”

---

## [Decision Letter · Decision Letter 1]

7 Sep 2023

PONE-D-23-22676R1Exome sequencing in 16 patients with pituitary stalk interruption syndrome: a monocentric studyPLOS ONE

Dear Dr. Brauner,

Thank you for submitting your manuscript to PLOS ONE. After careful consideration, we feel that it has merit but does not fully meet PLOS ONE’s publication criteria as it currently stands. Therefore, we invite you to submit a revised version of the manuscript that addresses the points raised during the review process.

We look forward to receiving your revised manuscript.

Kind regards,

Nejat Mahdieh

Academic Editor

PLOS ONE

Journal Requirements:

Reviewers' comments:

Reviewer's Responses to Questions

**Comments to the Author**

1. If the authors have adequately addressed your comments raised in a previous round of review and you feel that this manuscript is now acceptable for publication, you may indicate that here to bypass the “Comments to the Author” section, enter your conflict of interest statement in the “Confidential to Editor” section, and submit your "Accept" recommendation.

Reviewer #1: (No Response)

Reviewer #2: All comments have been addressed

2. Is the manuscript technically sound, and do the data support the conclusions?

Reviewer #1: Yes

Reviewer #2: Yes

3. Has the statistical analysis been performed appropriately and rigorously? 

Reviewer #1: N/A

Reviewer #2: Yes

4. Have the authors made all data underlying the findings in their manuscript fully available?

Reviewer #1: No

Reviewer #2: Yes

5. Is the manuscript presented in an intelligible fashion and written in standard English?

Reviewer #1: Yes

Reviewer #2: Yes

6. Review Comments to the Author

Reviewer #1: Given that PLOS ONE’s publication criteria do not include novelty, impact, or interest, the reviewer finds the study's conclusions to be well-founded. However, the reviewer identifies the following issues that need to be addressed.

Comment 1:

The diagnostic criteria for PSIS remain unclear. While the authors state that they have provided the diagnostic criteria for PSIS in the opening of the “Patient” section, they presented a clinical description of the patients rather than clear criteria.

A clearer presentation of the diagnostic criteria might look like: “A diagnosis of PSIS was made if the patient met at least ?two? of the following criteria: (1) an absent or ectopic posterior pituitary, (2) an interrupted, malformed, or thin pituitary stalk, and (3) a small anterior pituitary gland.” Based on Table 1 in the original version of the manuscript (which seems to be absent in the revised file), the patient 7 seems to have normal posterior pituitary gland, an interrupted pituitary stalk, and an anterior pituitary with a height of 4 mm. Is her anterior pituitary considered small? Please clarify the reference range for the height of the anterior pituitary gland.

Comment 2:

The reviewer was unable to locate the tables in the revised PDF version of the manuscript.

Reviewer #2: (No Response)

7. PLOS authors have the option to publish the peer review history of their article (what does this mean?). If published, this will include your full peer review and any attached files.

Reviewer #1: No

Reviewer #2: No

---

## [Author Response · Author response to Decision Letter 1]

22 Sep 2023

To Reviewer 1

Dear Reviewer,

Thank you for reviewing our manuscript. Please find joined the answers to your questions.

4. Have the authors made all data underlying the findings in their manuscript fully available?

Reviewer #1: No

Reviewer #2: Yes

Answer: All relevant data are within the manuscript and all novel variants are deposited with ClinVar.

6. Review Comments to the Author

Reviewer #1: Given that PLOS ONE’s publication criteria do not include novelty, impact, or interest, the reviewer finds the study's conclusions to be well-founded. However, the reviewer identifies the following issues that need to be addressed.

Comment 1:

The diagnostic criteria for PSIS remain unclear. While the authors state that they have provided the diagnostic criteria for PSIS in the opening of the “Patient” section, they presented a clinical description of the patients rather than clear criteria.

A clearer presentation of the diagnostic criteria might look like: “A diagnosis of PSIS was made if the patient met at least ?two? of the following criteria: (1) an absent or ectopic posterior pituitary, (2) an interrupted, malformed, or thin pituitary stalk, and (3) a small anterior pituitary gland.” Based on Table 1 in the original version of the manuscript (which seems to be absent in the revised file), the patient 7 seems to have normal posterior pituitary gland, an interrupted pituitary stalk, and an anterior pituitary with a height of 4 mm. Is her anterior pituitary considered small? Please clarify the reference range for the height of the anterior pituitary gland.

Answer: The diagnostic criteria of PSIS have been clarified as you suggested. A pituitary height of 4 mm is normal, as measured in 5 patients and this is indicated in the text as well as the reference with the normal range of the anterior pituitary height. 

Comment 2:

The reviewer was unable to locate the tables in the revised PDF version of the manuscript.

Answer: The two Tables are now included in the text. 

To Reviewer 2

Dear Reviewer,

Thank you for reviewing our manuscript.

Comment 2:

The reviewer was unable to locate the tables in the revised PDF version of the manuscript.

Answer: The two Tables are now included in the text.

---

## [Decision Letter · Decision Letter 2]

27 Sep 2023

Exome sequencing in 16 patients with pituitary stalk interruption syndrome: a monocentric study

PONE-D-23-22676R2

Dear Dr. Brauner,

We’re pleased to inform you that your manuscript has been judged scientifically suitable for publication and will be formally accepted for publication once it meets all outstanding technical requirements.

Kind regards,

Nejat Mahdieh

Academic Editor

PLOS ONE

Additional Editor Comments (optional):

Reviewers' comments:

Reviewer's Responses to Questions

**Comments to the Author**

1. If the authors have adequately addressed your comments raised in a previous round of review and you feel that this manuscript is now acceptable for publication, you may indicate that here to bypass the “Comments to the Author” section, enter your conflict of interest statement in the “Confidential to Editor” section, and submit your "Accept" recommendation.

Reviewer #1: All comments have been addressed

2. Is the manuscript technically sound, and do the data support the conclusions?

Reviewer #1: Yes

3. Has the statistical analysis been performed appropriately and rigorously? 

Reviewer #1: N/A

4. Have the authors made all data underlying the findings in their manuscript fully available?

Reviewer #1: Yes

5. Is the manuscript presented in an intelligible fashion and written in standard English?

Reviewer #1: Yes

6. Review Comments to the Author

Reviewer #1: The authors addressed all the comments, with the clear description of the diagnostic criteria and the inclusion of the missing tables in the main text.

One minor comment:

The authors might consider revising the definition of "N" in Table 1 from "normal puberty" to just "normal" because N is referred to as "N puberty" in the 6th column and also used in the "PP" and "Stalk" columns.

7. PLOS authors have the option to publish the peer review history of their article (what does this mean?). If published, this will include your full peer review and any attached files.

Reviewer #1: No

---

## [Editor Report · Acceptance letter]

5 Dec 2023

PONE-D-23-22676R2 

Exome sequencing in 16 patients with pituitary stalk interruption syndrome: a monocentric study 

Dear Dr. Brauner:

I'm pleased to inform you that your manuscript has been deemed suitable for publication in PLOS ONE. Congratulations! Your manuscript is now with our production department. 

Kind regards, 

on behalf of

Dr. Nejat Mahdieh 

Academic Editor

PLOS ONE